# Impact of varying accelerometer epoch length on physical activity patterns in adults: Considerations for public health

Rayane Haddadj[1,2]*, Charlotte Verdot[2], Benoît Salanave[2], Valérie Deschamps[2], Jérémy Vanhelst[3]

1 Department of Public Health and Nursing, Norwegian University of Science and Technology NTNU, Trondheim, Norway, 2 Nutritional Epidemiology Surveillance Team, Santé Publique France, Bobigny, France, 3 INSERM, INRAE, CNAM, Center of Research in Epidemiology and StatisticS (CRESS), Nutritional Epidemiology Research Team (EREN), Université Sorbonne Paris Nord and Université Paris Cité, Bobigny, France

* hdj.rayane@gmail.com

**Data Availability Statement:** The data are not deposited in publicly available repositories but are available on request from Santé publique France.

## Abstract

### Background

To process wearables sensors data, end-users face a wide variety of choices influencing physical activity (PA) patterns estimation. This study investigated the impact of varying epoch length on PA patterns in adults and World Health Organization (WHO) PA guidelines prevalence, assessed by accelerometer.

### Methods

The study included 181 adults (18–74 years) from the Esteban Study (2014–2016). Participants wore an accelerometer for 7 consecutive days. Data were processed with ActiLife® software using epoch lengths from 1 to 60 seconds to assess PA patterns. Difference of PA patterns between epoch length was assessed using repeated measures ANOVA. Difference in meeting WHO PA guidelines was assesses using Fisher's exact test.

### Results

Significant changes were observed in sedentary behaviour and PA intensities with epoch length variation (p<0.001). Longer epochs led to reduced moderate and vigorous PA, increased light PA, and less sedentary time, affecting adherence to WHO PA guidelines.

### Conclusion

Result underline the importance of careful epoch length selection when processing accelerometer data to accurately assess sedentary behaviour and PA in adults. Shorter epochs seem preferable to capture short and spontaneous PA bouts and preventing underestimation of MVPA along with prolonged PA bouts. Further investigation including a PA reference measure is needed to confirm these findings and their implications for adult health.

For more information, please consult: https://www.santepubliquefrance.fr/les-actualites/2019/appel-a-projet-d-ouverture-des-bases-de-donnees-y-compris-de-la-collection-biologique-de-l-etude-esteban-2014-2016.

**Funding:** The author(s) received no specific funding for this work.

**Competing interests:** The authors have declared that no competing interests exist.

## Introduction

Previous studies have shown that an active lifestyle is beneficial for biological, psychological and social dimensions of health [1]. Conversely, sedentary behaviours (SB) are associated with poor health and an increased risk to develop various chronic diseases [1]. Obtaining the most precise physical activity (PA) measure is a major challenge to correctly define the population's PA levels and to better understand the relationship between PA and various health outcomes. PA is mostly assessed using questionnaire because of their several strengths (e.g., low cost, non-invasive, easy to administrate, to score and to interpret) [2, 3]. However, due to inherent weaknesses using self-reports to assess PA (e.g., memory bias, social desirability, difficulty to assess light PA) [3, 4], devices like accelerometer were developed and are now widely used in various large scale epidemiological studies [5–7]. Accelerometers represent an interesting–non-invasive–alternative to self-reports because they erased participants subjectivity. However, existence of a variety of data collection protocols (e.g., wear location and device brand) and data processing approach (e.g., raw data or proprietary counts) can be confusing for investigators [8]. Heterogeneity in methods prevents researchers from complete comparability between studies. Thus, a collective effort is needed to produce knowledge that help researchers and accelerometer users to select appropriate methods when collecting and processing accelerometer data.

Epoch length is a major concern regarding accelerometer data processing. It refers to the amount of time over which a movement data is summed and stored [9]. Current generation of accelerometers enables researchers to record and store raw accelerations. When downloading data, proprietary software and open-source packages usually propose to sum acceleration values recorded over a selected interval or epoch. Subsequently, each epoch can be classified regarding PA intensities (sedentary, light, moderate, vigorous) using existing thresholds [8]. Previous studies have shown that epoch length affects PA patterns [9–11] and associations between PA and health outcomes [12, 13]. Those evidence support that epoch length represents a key factor that needs to be understood and standardized across studies. Previous studies only focused on children and data are unavailable for general adult population using "GT3X/+" generation of accelerometer, the most broadly used tool in research [8, 14]. A study have also demonstrated significant variations in PA accumulation patterns between adults and children [15]. Therefore, it remains unclear how well data processing recommendations established using children's data can be applied to adults. To date, no study has assessed the impact of varying epoch lengths on the prevalence on PA levels in free-living conditions.

The primary aim of the study was to assess the effect of varying epoch length on time spent in SB and various PA intensities in French adults. The secondary aim was to determine the prevalence of meeting World Health Organization (WHO) PA guidelines according to epochs lengths used.

## Methods

### Study design

The present study was performed under the framework of the Esteban study (2014–2016) [16]. The Esteban study, conducted by Santé publique France, was a nationwide cross-sectional study that aimed to assess nutritional status, PA, SB, chronic diseases prevalence and environmental exposure level among French population. Details of the recruitment, sampling, standardization, and harmonization processes were published elsewhere [16].

Written informed consent was obtained from the participants. The Esteban study was approved by the ethics committee and the Advisory Committee on Information Treatment in

the field of Health Research (CCTIRS). All procedures were performed in accordance with the ethical standards of the Helsinki Declaration of 1975 as revised in 2008. Data were extracted for research purposes from the Esteban database in March 2023.

In the Esteban study, 230 adults took part in the accelerometry protocol. Of these, 49 participants were excluded (13 did not meet wear criteria, 23 had unreadable files, and 13 had missing age information preventing from selecting appropriate threshold; **Fig 1**).

## Participants characteristics

All participants were adults aged 18 to 74 years old and living in France. Weight and height were measured by a health professional using standard procedures and equipment as part of the health examination. Body mass index (BMI) was calculated as weight/height$^2$ (kg/m$^2$).

## Physical activity

Participants wore a triaxial accelerometer (ActiGraph$^®$, wGT3X-BT model, Pensacola, FL, USA) which is a small (4,6x3,3x1,5 cm) and light (19 grams) device that has been validated against oxygen consumption [17]. Accelerometer was worn during 7 consecutive days on the right hip, attached with an elastic belt. Participants were asked to not wear accelerometer during sleep or water activities and to follow their regular daily routine.

Raw accelerometer data were downloaded and analysed according to several epoch lengths (i.e., 1, 5, 10, 15, 30 and 60 seconds) using ActiLife$^®$ software (v6.11.9, Pensacola, FL, USA). Data from the three axes were combined and expressed as vector magnitude, calculated as the square root of the sum of squared activity counts from each axis. Participants were excluded from analyses if they did not provide at least 3 days of recordings with $\geq 8$ hours of activity per day. Non-wear time was defined as 60 minutes of consecutives zeros and Choi *et al.* algorithm for adults aged 18–54 and 55–74 years old, respectively [8, 18].

To classify SB and PA intensities based on age, several thresholds were applied to vector magnitude counts per minute (cpm). For participants aged 18–54 years old, sedentary time was defined as <200 cpm, light PA (LPA) as 200–2689 cpm, moderate PA (MPA) as 2690–6166 cpm, and vigorous PA (VPA) as $\geq 6167$ cpm [19]. For participants aged 55–74 years old, sedentary time was defined as <200 cpm, LPA as 200–2751 cpm, MPA as 2752–9159 cpm, and VPA as $\geq 9160$ cpm [17]. For epoch lengths other than 60 seconds, threshold values were adjusted proportionally to match the selected epoch length.

Daily time spent in SB and different PA intensities was computed as the average time spent in each intensity during valid days. Achieving WHO PA guidelines meant for adults carrying out a least 150–300 minutes of moderate-intensity PA (MPA) per week, or 75–150 minutes of VPA or an equivalent combination of both, accordingly to 2020 WHO PA guidelines [1]. The formula: (ΣMPA on valid days + 2*ΣVPA on valid days)/number of valid days*7 $\geq$ 150 min was used.

## Statistical analysis

Data were processed using Stata$^®$ software (version 14.2, Texas, USA). Mean and standard deviation were presented for quantitative variables and proportions were described for qualitative variables. Repeated measures analysis of variance (ANOVA) was performed to define effect of varying epoch length on SB and PA intensities and wear time. Post hoc pairwise t-tests, with Bonferroni adjustment for multiple comparisons, were computed in case of a significant overall ANOVA. Effect of epoch length on meeting WHO PA guidelines was defined using Fisher's exact test. Statistical significance was defined for $p < 0.05$ and adjusted for multiple comparisons.

**Fig 1. Flowchart of inclusion (accelerometry protocol) in Esteban study (2014–2016).**

**Table 1. Characteristics of participants (n = 181).**

|  | All | Men (n = 80) | Women (n = 101) | *P*\* |
|---|---|---|---|---|
| Age (years; %) |  |  |  |  |
| 18–39 | 16.6 | 20.0 | 13.9 | 0.17 |
| 40–54 | 33.1 | 37.5 | 29.7 |  |
| 55–74 | 50.3 | 42.5 | 56.4 |  |
| Body mass index (kg.m$^{-2}$) | 26.2 ± 4.7 | 25.9 ± 3.3 | 26.4 ± 5.5 | 0.53 |
| Normal weight (%) | 43.1 | 38.2 | 46.7 | 0.28 |
| Overweight and obesity (%) | 56.9 | 61.8 | 53.3 |  |

\**P* value show results from Chi-square test between gender for age and normal weight; and from two-sample t-test between gender for BMI

## Results

The physical characteristics of the 181 participants included in the present study are described in **Table 1**.

**Table 2** displays wear time, SB time, and time spent in different PA intensities according to various epoch lengths. There is no significant effect of epoch length on wear time ($p = 0.34$). In contrast, significant differences were found for the time spent in SB, LPA, MPA, VPA, and moderate-to-vigorous PA (MVPA) according to epoch length used ($p < 0.001$). Time spent in SB decreased by up to 28.7% when shifting from 1-second epoch to 60-seconds epoch, while MPA, VPA, and MVPA decreased by up to 50.6%, 77.6%, and 53.6%, respectively, with longer epoch lengths. LPA showed an increase of up to 177.2% when shifting from 1-second epoch to 60-seconds epoch. Bonferroni adjusted post-hoc pairwise t-tests revealed significant differences in time classified in SB and PA intensities for each epoch comparison. Wear time was not affected by epoch length selection ($p = 0.34$). Further stratification on sex, age category and BMI status did not change the results (S1-S7 Tables in S1 File).

Increasing epoch length was significantly associated with a lower prevalence of meeting the WHO PA guidelines, ranging from 99.4% for 1-second epoch to 12.2% for 60-second epoch (**Fig 2**; $p < 0.001$).

**Table 2. Daily wear time and time spent in each physical activity intensity using different epoch lengths.**

| Epoch length | Wear time | SB | LPA | MPA | VPA | MVPA |
|---|---|---|---|---|---|---|
| 1-sec | 877.5 ± 93.5 | 648.3 ± 100.0[b,c,d,e,f] | 133.7 ± 40.7[b,c,d,e,f] | 84.8 ± 36.1[b,c,d,e,f] | 10.7 ± 14.8[b,c,d,e,f] | 95.5 ± 42.6[b,c,d,e,f] |
| 5-sec | 876.7 ± 92.6 | 581.9 ± 104.6[c,d,e,f] | 213.9 ± 61.0[c,d,e,f] | 75.3 ± 36.9[c,d,e,f] | 5.6 ± 12.4[c,d,e,f] | 80.9 ± 42.4[c,d,e,f] |
| 10-sec | 876.8 ± 92.6 | 548.9 ± 107.8[d,e,f] | 256.1 ± 70.8[d,e,f] | 67.7 ± 37.1[d,e,f] | 4.1 ± 11.3[e,f] | 71.8 ± 42.2[d,e,f] |
| 15-sec | 876.9 ± 92.6 | 528.6 ± 109.5[e,f] | 283.0 ± 77.0[e,f] | 61.8 ± 36.7[e,f] | 3.5 ± 10.7[f] | 65.3 ± 41.7[e,f] |
| 30-sec | 877.1 ± 92.6 | 495.2 ± 112.2[f] | 327.7 ± 86.6[f] | 51.3 ± 35.5[f] | 2.8 ± 9.6 | 54.1 ± 40.4[f] |
| 60-sec | 877.4 ± 92.5 | 462.5 ± 114.4 | 370.6 ± 95.1 | 41.9 ± 34.1 | 2.4 ± 8.5 | 44.3 ± 38.7 |
| *P* value\* | 0.34 | < 0.001 | < 0.001 | < 0.001 | < 0.001 | < 0.001 |

Abbreviations: SB: sedentary behaviour, LPA: light physical activity; MPA: moderate physical activity; VPA: vigorous physical activity; MVPA: moderate to vigorous physical activity

All values are expressed in mean minutes ± SD

\*Repeated measures ANOVA. Pairwise post-hoc comparisons were performed using t-test, after applying a Bonferroni correction for multiple comparisons (a = 1-sec, b = 5-sec, c = 10-sec, d = 15-sec. e = 30-sec, f = 60-sec)

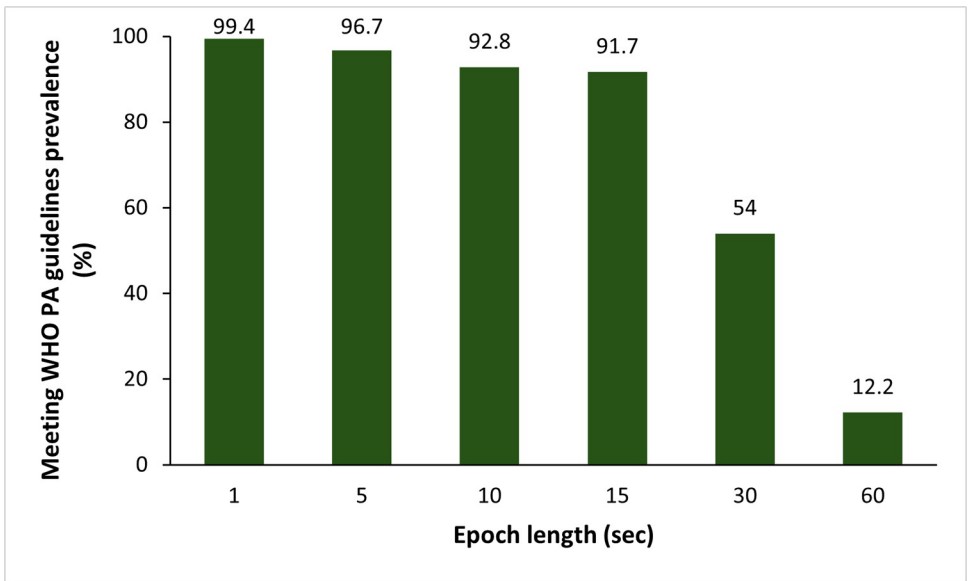

**Fig 2. Meeting World Health Organization physical activity guidelines prevalence according to epoch lengths (n = 181).**

## Discussion

Our study aimed to assess the effect of varying epoch length on PA patterns in adults. Findings suggest that longer epoch lengths are associated with a lower time spent in SB, MPA, VPA, MVPA and a higher time spent in LPA. This inevitably has an impact on the prevalence of meeting WHO PA guidelines, with selection of longer epoch lengths resulting in a lower prevalence of meeting PA guidelines. Furthermore, pairwise comparisons showed significant differences in time classified in SB and PA intensities across different epoch lengths, indicating poor comparability between them.

Only time classified as LPA increased with longer epoch lengths. This result suggests that LPA bouts occurring within SB or within MPA bouts may contribute to this increase through an averaging process as described previously [20]. Previous research conducted in post-menopausal women with overweight using ActiGraph GT1M reported that the choice of epoch length affected the time spent in SB, LPA, MPA and VPA, finding results similar to those observed in our study for each intensity [13]. Conversely, Ayabe *et al.* found that MPA and MVPA could alternatively decrease then increase with longer epoch lengths [21]. However, this study collected data using a different model of accelerometer (Lifecorder, Susuken[®], Nagoya, Japan) and examined different epoch lengths compared to our study (i.e. 4, 20 and 60s), which could potentially account for discrepancies observed in results. Previous studies carried out in children and adolescents have also investigated the influence of epoch length on time spent in different PA intensities, yielding varying results. While epoch length consistently affected time spent in SB, LPA and VPA similarly, it displayed distinct effects on MPA and MVPA across studies [11, 22–25]. These differences could potentially be explained by variations in PA accumulation patterns between children and adults. Children tend to engage in more short and sporadic periods of MVPA throughout the day [15, 26]. Consequently, since short VPA bouts within longer MPA bouts are more common in children, this could potentially result in a greater amplification of MPA at the expense of VPA when longer epoch lengths are selected for children. However, no previous study has attempted to understand

whether varying epoch lengths affect children and adults differently. Further research is needed to address this gap.

This study is the first to assess the impact of epoch length choice on meeting PA guidelines prevalence in adults using GT3X/+ accelerometer. Our analysis indicates that a higher epoch length is associated with a lower prevalence of meeting WHO PA guidelines. These results are clearly associated with changes in epoch length, reflecting the consistent decrease of MPA and VPA at the expense of LPA as epoch length increases. Previous research conducted in children and adolescents interestingly concluded that epoch length had no significant impact on meeting PA guidelines. This conclusion was drawn from data indicating that only VPA was significantly "misclassified" as MPA with longer epoch lengths [20, 23]. According to these results, adherence to PA guidelines in children was not influenced by epoch length choice, as meeting guidelines is assessed by combining time spent in MPA and VPA. Additionally, in adults, WHO PA guidelines consider MPA and VPA separately in the calculation, which logically explains the observed decrease in prevalence as both MPA and VPA decrease with longer epoch lengths.

Results show that epoch length choice needs to be considered when processing adults data whether for PA classification or defining meeting guidelines prevalence purposes. A longer epoch length continuously increases time classified as LPA and decreases time classified as SB, MPA, VPA and MVPA. Even if adults tend to accumulate PA with less short and spontaneous bouts than children, it seems that shorter epoch lengths should be preferred to capture short bouts of activity occurring in daily life (e.g. quick stairs climbing, short run, get up to fetch something) and to avoid SB and MPA minimization due to some averaging process. Furthermore, variations observed between epoch lengths regarding PA classification and meeting WHO PA guidelines prevalence indicate that studies utilizing different epoch lengths should not be compared. However, as this is the first study in adults from general population using GT3X accelerometer, results must be interpreted cautiously and further research are needed to better understand to which extent epoch length selection affects PA patterns in adults. Moreover, the processed data volume is relatively low compared to the main cohort study population due to the limited number of accelerometers and investigators available.

## Strength and limitations

The main strength of the present study is the use of multiple epoch lengths, which provides a broad overview of classification variations across settings. In addition, the use of a relatively large sample of data collected in free-living conditions, which certainly enabled us to capture a wide variety of PA types and patterns that might be challenging to recreate or capture in closed-environment protocols. The limitations of this study include the absence of a reference measure for PA, which prevents us from determining which epoch length accurately captures PA among participants. Moreover, the results may differ if different accelerometer body locations or alternative cut-off points had been used. Future research using direct observation, as has been done previously in children [9], could provide a valuable point of comparison.

## Conclusion

This study found that the choice of epoch lengths is associated with variations in PA patterns and the prevalence of meeting WHO PA guidelines in adults. These findings have implications for public health and research considerations related to data processing. Based on our analysis, it appears that using shorter epoch lengths allows to better capture VPA and prevent an averaging process that results in increased LPA. Future research is needed to confirm these

findings, define more precisely which epoch to select, and assess the association between epoch length choice and various health outcomes in adults.

## Supporting information

**S1 File. Daily wear time and time spent in each physical activity intensity using different epoch lengths stratified on sex, age and body mass index.**
(DOCX)

## Author Contributions

**Conceptualization:** Rayane Haddadj.

**Formal analysis:** Rayane Haddadj.

**Investigation:** Rayane Haddadj.

**Methodology:** Rayane Haddadj, Charlotte Verdot, Benoît Salanave, Jérémy Vanhelst.

**Resources:** Charlotte Verdot, Benoît Salanave, Valérie Deschamps.

**Software:** Rayane Haddadj, Benoît Salanave.

**Supervision:** Charlotte Verdot, Jérémy Vanhelst.

**Validation:** Charlotte Verdot, Benoît Salanave, Valérie Deschamps, Jérémy Vanhelst.

**Writing – original draft:** Rayane Haddadj.

**Writing – review & editing:** Rayane Haddadj, Charlotte Verdot, Benoît Salanave, Valérie Deschamps, Jérémy Vanhelst.

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
