## [Decision Letter · Decision Letter 0]

29 Sep 2024

PONE-D-24-20121Impact of varying accelerometer epoch length on physical activity patterns in adults: considerations for public healthPLOS ONE

Dear Dr. Haddadj,

Thank you for submitting your manuscript to PLOS ONE. After careful consideration, we feel that it has merit but does not fully meet PLOS ONE’s publication criteria as it currently stands. Therefore, we invite you to submit a revised version of the manuscript that addresses the points raised during the review process.

We look forward to receiving your revised manuscript.

Kind regards,

Julio Alejandro Henriques Castro da Costa

Academic Editor

PLOS ONE

Journal Requirements:

Reviewers' comments:

Reviewer's Responses to Questions

**Comments to the Author**

1. Is the manuscript technically sound, and do the data support the conclusions?

Reviewer #1: Yes

2. Has the statistical analysis been performed appropriately and rigorously? 

Reviewer #1: I Don't Know

3. Have the authors made all data underlying the findings in their manuscript fully available?

Reviewer #1: Yes

4. Is the manuscript presented in an intelligible fashion and written in standard English?

Reviewer #1: Yes

5. Review Comments to the Author

Reviewer #1: It was a pleasure to review this well-written manuscript, which examined the impact of varying epoch lengths on accelerometer-derived time spent in activity behaviours in adults. The findings can be of interest and advance knowledge of those in the activity behaviour research field. I have provided some comments/questions/suggestions below on the manuscript that the authors may consider if they find them relevant to improve the scientific quality.

Major comments:

1. Several cut-off points for the accelerometer-derived time spent in the different physical activity spectrum have been reported in the literature, and this is partly determined by several factors including the age distributions of the studied participants. The authors could help the reader by defining the specific cut-off thresholds for the different activity behaviours (sedentary behaviour and the different intensities of physical activity).

2. Given the wide age range of the included participants, the authors could consider providing a brief justification of the 200 counts per minute (cpm) threshold for sedentary behaviour to help orient the reader.

3. Another important factor which has been noted in the literature regarding activity behaviours (sedentary behaviour, and different intensities of physical activity) cut-off points selection for accelerometer raw data processing is the epoch length. The authors could clarify the cut-off thresholds applied for the various epoch lengths for the activity behaviours.

4. Did the authors consider checking or accounting for the potential influence of age distribution, gender and weight/BMI distribution on the estimated time spent in the activity behaviours for the various epoch lengths?

Minor comments:

1. The authors mentioned in the abstract results that “Longer epochs led to reduced moderate and vigorous PA, increased light PA, and less sedentary time, affecting adherence to World Health Organization PA guidelines” (lines 23 – 25). But it is clear in the abstract if that was examined.

2. The authors could be a little specific about the 13 participants excluded based on sociodemographic characteristics (…lines 83 – 84). Considering the nature of the analysis reported in the study, are the missing variables relevant for processing the accelerometer data?

3. There may probably be a typo in the description provided on …lines 113 – 115 regarding “qualitative”.

4. It may interest some readers if the pair-wise post hoc test results are also reported and interpreted as it may provide some relevant insights regarding epoch length selection.

5. The authors could also consider discussing their observations within the context of cut-off points used for the data processing.

6. It may interest some readers if the authors also note in the discussion the potential influence of the body location where the participant wore the device on the estimation of the time spent in the activity behaviours.

6. PLOS authors have the option to publish the peer review history of their article (what does this mean?). If published, this will include your full peer review and any attached files.

Reviewer #1: No

---

## [Author Response · Author response to Decision Letter 0]

13 Nov 2024

Dear Editor, 

Thank you for these comments designed to improve our paper, “Impact of varying accelerometer epoch length on physical activity patterns in adults: considerations for public health” which we have addressed below. We greatly appreciate the time and effort put forth by reviewer to improve our paper. We hope our manuscript will now be suitable for publication in Plos One. 

Sincerely,

Rayane Haddadj

Major comments

1. Several cut-off points for the accelerometer-derived time spent in the different physical activity spectrum have been reported in the literature, and this is partly determined by several factors including the age distributions of the studied participants. The authors could help the reader by defining the specific cut-off thresholds for the different activity behaviours (sedentary behaviour and the different intensities of physical activity).

Thank you for this remark. We agree that providing the specific thresholds for the different activity behaviours would enhance the clarity of the manuscript. In response, we have added the exact accelerometer-derived thresholds used to classify sedentary behaviour and the various intensities of physical activity (light, moderate, and vigorous). These thresholds are now clearly defined, considering the age distribution of our study population, as supported by the relevant literature (lines 110 to 116).

2. Given the wide age range of the included participants, the authors could consider providing a brief justification of the 200 counts per minute (cpm) threshold for sedentary behaviour to help orient the reader.

We acknowledge the importance of justifying the 200 counts per minute (cpm) threshold for sedentary behaviour. To address this, we have included a relevant reference in the manuscript that supports the use of this threshold across different age groups (lines 112 and 114). 

3. Another important factor which has been noted in the literature regarding activity behaviours (sedentary behaviour, and different intensities of physical activity) cut-off points selection for accelerometer raw data processing is the epoch length. The authors could clarify the cut-off thresholds applied for the various epoch lengths for the activity behaviours.

We now have clarified how the thresholds for sedentary behaviour and physical activity were adjusted according to the different epoch lengths used for accelerometer data processing (lines 100 and 101).

4. Did the authors consider checking or accounting for the potential influence of age distribution, gender and weight/BMI distribution on the estimated time spent in the activity behaviours for the various epoch lengths?

Thank you for this interesting suggestion. We have run additional analysis stratified on sex, age category and BMI status (Supplement files). We also commented those results in the text (line 149 and 150).

Minor comments

1. The authors mentioned in the abstract results that “Longer epochs led to reduced moderate and vigorous PA, increased light PA, and less sedentary time, affecting adherence to World Health Organization PA guidelines” (lines 23 – 25). But it is clear in the abstract if that was examined.

Thank you for this remark. To clarify this point, we have added a secondary objective in the abstract that states our examination of the effects of epoch length selection on computed prevalence of WHO physical activity guidelines (line 16 and 17).

2. The authors could be a little specific about the 13 participants excluded based on sociodemographic characteristics (…lines 83 – 84). Considering the nature of the analysis reported in the study, are the missing variables relevant for processing the accelerometer data?

We have now clarified that the 13 participants were excluded due to missing age data, which is critical for appropriately selecting the accelerometer thresholds (line 87, 88 and Figure 1).

3. There may probably be a typo in the description provided on …lines 113 – 115 regarding “qualitative”.

Thank you for pointing this out. We have corrected the typo by changing "qualitative" to "quantitative".

4. It may interest some readers if the pair-wise post hoc test results are also reported and interpreted as it may provide some relevant insights regarding epoch length selection.

We have added the pair-wise post hoc test results, in Table 2 and presented them in the results (lines 147 to 149) and discussion (line 184 to 186).

5. The authors could also consider discussing their observations within the context of cut-off points used for the data processing.

We have added a sentence in the limitations to highlight the context of cut-off points (line 241 and 242).

6. It may interest some readers if the authors also note in the discussion the potential influence of the body location where the participant wore the device on the estimation of the time spent in the activity behaviours.

Thank you for this interesting remark. We have added a comment on this topic in the limitations section (line 241and 242).

---

## [Decision Letter · Decision Letter 1]

9 Dec 2024

Impact of varying accelerometer epoch length on physical activity patterns in adults: considerations for public health

PONE-D-24-20121R1

Dear Dr. Haddadj,

We’re pleased to inform you that your manuscript has been judged scientifically suitable for publication and will be formally accepted for publication once it meets all outstanding technical requirements.

Kind regards,

Julio Alejandro Henriques Castro da Costa

Academic Editor

PLOS ONE

Additional Editor Comments (optional):

Reviewers' comments:

Reviewer's Responses to Questions

**Comments to the Author**

1. If the authors have adequately addressed your comments raised in a previous round of review and you feel that this manuscript is now acceptable for publication, you may indicate that here to bypass the “Comments to the Author” section, enter your conflict of interest statement in the “Confidential to Editor” section, and submit your "Accept" recommendation.

Reviewer #2: (No Response)

Reviewer #3: All comments have been addressed

2. Is the manuscript technically sound, and do the data support the conclusions?

Reviewer #2: Yes

Reviewer #3: Yes

3. Has the statistical analysis been performed appropriately and rigorously? 

Reviewer #2: Yes

Reviewer #3: Yes

4. Have the authors made all data underlying the findings in their manuscript fully available?

Reviewer #2: Yes

Reviewer #3: Yes

5. Is the manuscript presented in an intelligible fashion and written in standard English?

Reviewer #2: Yes

Reviewer #3: Yes

6. Review Comments to the Author

Reviewer #2: Some results are repeated twice. See L130: There is no significant effect of epoch length on wear time (p=0.34) and L137-138: Wear time was not affected by epoch length selection (p=0.34)

Reviewer #3: We thank the authors for their detailed responses to the review comments and revision work. After review, the authors have fully responded to all major and minor review comments and improved the methodology, results, discussion and figures accordingly. The revised manuscript is clearly structured, rigorously argued, with reasonable data analyses, and the limitations of the study and future research directions have been appropriately discussed.

Overall, the article has met the criteria for publication and no further revisions have been suggested.

7. PLOS authors have the option to publish the peer review history of their article (what does this mean?). If published, this will include your full peer review and any attached files.

Reviewer #2: No

Reviewer #3: No

---

## [Editor Report · Acceptance letter]

17 Dec 2024

PONE-D-24-20121R1 

PLOS ONE

Dear Dr. Haddadj, 

I'm pleased to inform you that your manuscript has been deemed suitable for publication in PLOS ONE. Congratulations! Your manuscript is now being handed over to our production team.

Kind regards, 

on behalf of

Dr. Julio Alejandro Henriques Castro da Costa 

Academic Editor

PLOS ONE